# Influence of the Titanium Implant Surface Treatment on the Surface Roughness and Chemical Composition

**DOI:** 10.3390/ma13020314

**Published:** 2020-01-09

**Authors:** Ana Isabel Nicolas-Silvente, Eugenio Velasco-Ortega, Ivan Ortiz-Garcia, Loreto Monsalve-Guil, Javier Gil, Alvaro Jimenez-Guerra

**Affiliations:** 1Associate Professor of Restorative Dentistry, Professor of Master in Mucogingival, Periodontal and Implant Surgery, School of Dentistry, University of Murcia, 30008 Murcia, Spain; ainicolas@um.es; 2Professor of Comprehensive Dentistry for Adults, Director of Master in Implant Dentistry, Faculty of Dentistry, University of Seville, 41009 Sevilla, Spain; 3Associate Professor of Comprehensive Dentistry for Adults, Professor of Master in Implant Dentistry, Faculty of Dentistry, University of Seville, 41009 Sevilla, Spain; ivanortizgarcia1000@hotmail.com (I.O.-G.); lomonsalve@hotmail.es (L.M.-G.); alopajanosas@hotmail.com (A.J.-G.); 4Chairman of Bioengineering Institute of Technology, Universitat Internacional de Catalunya, 08017 Barcelona, Spain; xavier.gil@uic.es

**Keywords:** surface roughness, surface chemical composition, dental implant contamination, surface treatment technique

## Abstract

The implant surface features affect the osseointegration process. Different surface treatment methods have been applied to improve the surface topography and properties. Trace of different elements may appear on the implant surface, which can modify surface properties and may affect the body’s response. The aim was to evaluate the roughness based on the surface treatment received and the amount and type of trace elements found. Ninety implants (nine different surface treatment) were evaluated. Roughness parameters were measured using white-light-interferometry (WLI). The arithmetical mean for *R*_a_, *R*_q_, *R*_t_, and *R*_z_ of each implant system was calculated, and Fisher’s exact test was applied, obtaining *R*_a_ values between 0.79 and 2.89 µm. Surface chemical composition was evaluated using X-ray photoelectron spectroscopy (XPS) at two times: as received by the manufacturer (AR) and after sputter-cleaning (SC). Traces of several elements were found in all groups, decreasing in favor of the Ti concentration after the sputter-cleaning. Within the limitations of this study, we can conclude that the surface treatment influences the roughness and the average percentage of the trace elements on the implant surface. The cleaning process at the implant surface should be improved by the manufacturer before assembling the implant.

## 1. Introduction

Implant surface characteristics have been shown to play an essential role in the osseointegration process [1]. The cellular responses depend upon the chemical and physical characteristics of the substrate [2] and particularly upon its particle size [3], crystallinity [4,5], chemical composition [6], and surface structure [7]. The attachment capability by human stromal cells to smooth titanium surface is low [8] and can lead to the formation of fibrous tissue layer between the implant and the surrounding bone [9]. To increase biocompatibility and cell viability, modifications affecting topography, roughness surface characteristics, and chemical surface composition must be done [10].

Regarding the implant surface roughness, the osteoblast activity can be seen increased with micro-rough from 1 to 100 µm when it is compared to untreated or smooth surfaces [11]. The most commonly used methods to acquire these roughness characteristics are sandblasting with different metals or metal oxides, etching, machining, micron-sized metal bead coatings, or anodization. Most of the current commercial implant systems have a height variation (*R*_a_) ranging from 1 to 2 µm. The microtopographic features of the implant surface (peaks, valleys, and protrusions) are an essential factor in the biological response and the configuration of the bone-implant interface [12,13]. The implant surface topography can be classified as smooth (*R*_a_ < 0.5 µm), minimally rough (*R*_a_ 0.5–1.0 µm), moderately rough (*R*_a_ 1.0–2.0 µm), and highly rough (*R*_a_ > 2.0 µm) [12]. Surface properties are a critical factor for achieving clinical success [14,15].

On the opposite, despite the increasing interest in performing new surfaces that allow quick osseointegration, the interactions with soft tissues are still less investigated [16,17]. When the implant surface is treated by any of the methods exposed above, traces of the materials used, such as metals, metal ions, lubricants, and detergents, may appear. These elements can modify surface properties even when present in small quantities and may affect the body´s response during the osseointegration process, leading to the formation of undesirable tissues between bone/implant interface [18,19]. Although the effects caused by these low concentration trace elements are relatively weak studied, there is a broad agreement that comprehensive control of the implant surface and elimination of undesirable chemical compounds must be done to improve implant quality [20].

The presence of organic contamination (carbon) on all implant surfaces cannot be avoided since the hydrocarbons present in the atmosphere are almost instantly adsorbed on the titanium surface exposed to the air. Nevertheless, the presence of other elements, such as sodium, chlorine, calcium, sulfur, or silicon indicates that these impurities have not been removed by the cleaning process. This may occur due to the roughness itself; hidden areas remain where the ionic beam cannot reach to carry out the cleaning process [21]. The presence of some of these trace elements may even be adequate, such as calcium phosphate, which induces the formation of biochemical junctions that facilitate the quick and intense osseointegration of the implant, especially in the early stages of bone healing. In this sense, calcium phosphate has been documented as biocompatible with osteoconductive properties [22].

Most of the trace elements found do not have these favorable properties, and they can even alter cell viability. The presence of deoxyribonucleic acid (DNA) and lactate dehydrogenase (LDH) assess nuclear and cell membrane alterations. This presence is usually low, confirming the safety of implants for clinical use, but the identification of cytoplasmatic cell components could demonstrate some degree of toxicity on some implant surfaces [23]. This toxicity, together with the ionic release, could somehow affect the osseointegration process and perhaps could be the precursor of a future peri-implant disease [23].

Currently, there are different methods used by the manufacturer before the final implant assembly to clean the trace elements deposited on the implant surface. Some of these methods are sputter cleaning [24], which is a physical technique to remove surface layers, material is physically removed from a surface by bombardment of ions generated in a plasma, being the gas used generally inert or nonreactive (argon or helium more frecuently), the goal of ion beam sputtering is to remove unwanted layers without damaging the rest and it is widely used before surface analysis; abrasive air [25]; laser or photodynamic therapy [26] between others.

Therefore, the goal of the present study was to evaluate the relationship between the roughness created in different implant systems based on their surface treatment and the amount and type of trace elements found on their surface.

## 2. Materials and Methods

For the development of this experimental study, the surface of different implant systems was analyzed. The surface roughness and the chemical elements surface composition was measured for each implant as received by the manufacturer and after a cleaning process.

### 2.1. Dental Implants and Experimental Groups

Ninety implants distributed in nine implant systems with different surface treatments were evaluated (*n* = 10 per group; Table 1).

Group I: Straumann^®^ Implant System: contained 10 Straumann^®^ Bone Level (BL; Straumann) with cylindrical (parallel) outer contour with a SLA^®^ surface treatment, which is based on a large-grit sandblasting technique that generates the macro-roughness, followed by acid-etching that superposes a micro-roughness.

Group II: Microdent^®^ Implant System: contained 10 Microdent Genius^®^ (Microdent Implant System) conical connection dental implants with a surface treatment, which is made by applying a physical attack with abrasive alumina particles (sandblasting) at high pressure. This method is registered as Abrasive Treatment Extreme Cleaning^®^ (ATEC^®^). The implant is sandblasted along its entire length.

Group III: Astra Tech^®^ Implant System: contained 10 Astra Tech OsseoSpeed^®^ (Dentsply-Sirona) tapered implants, which surface treatment is made by a titanium dioxide sandblasting at high pressure, followed by a fluoride nanostructure treatment. 

Group IV: Avinent^®^ Implant System: contained 10 Ocean^®^ (Avinent), which Biomimetic^®^ surface is treated by sandblasting at high pressure, followed by the addition of calcium and phosphorus. 

Group V: Biomet 3i^®^ Implant System: contained 10 Osseotite^®^ (Zimmer Biomet), which surface is treated by a double acid-etching technique. 

Group VI: Klockner^®^ Implant System: contained 10 Vega^®^ (Klockner), which surface is treated by a two-phase technique starting with an alumina particle attack followed by a thermochemical treatment (alkaline immersion plus heat treatment). 

Group VII: Mozograu-Ticare^®^ Implant System: contained 10 inHex^®^ (Ticare) with RBM TC^®^ surface, obtained by applying a physical attack with resorbable particles (sandblasting) at high pressure followed by a double acid-etching. 

Group VIII: Nobel Biocare^®^ Implant System: contained 10 NobelReplace^®^ tapered (Nobel Biocare) with TiUnite^®^ surface characterized by a thick and moderately rough titanium oxide layer with a high degree of crystallinity. 

Group IX: Galimplant^®^ Implant System: contained 10 IPX^®^ (Galimplant) with Nanoblast^®^ surface obtained after abrasive sandblasting at high pressure followed by a triple acid-etching technique.

### 2.2. Evaluation of the Implant Surface Roughness

Implant surface roughness was evaluated for all the implants included in this experimental study. Different roughness parameters were recorded.

#### 2.2.1. Data Acquisition

For the quantitative evaluation of the surface roughness of the nine different implant systems, a last generation white light interferometry (WLI) equipment (Optical Profiling System, Wyko NT9300, Veeco Instruments, New York, NY, USA) was used. WLI is a non-contact optical method that allows measurements on 3D structures by using a wave superposition principle with a visible-wavelength light (white light).

Measurements were made using the white light optical interferometry (WLOI) non-contact topography characterization technique with a vertical scanning interferometry mode (VSI). A magnification of 20× with a 1× field of view (FOV) was used, obtaining an image size of 227 × 298 µm^2^. Nine areas were randomly selected on the middle part of the implant surface, and the average of each parameter evaluated was calculated with Wyko Vision 232TM Software (Veeco Instruments, New York, NY, USA). The following image filtering was set to calculate roughness parameters: 1. Missing data removal; 2. height at level “strong” to eliminate excessively high peaks that may appear by image artifact; and 3. surface shape selecting “waveform removal” to eliminate the influence of the slight cylindrical curvature of the selected areas and turn the curved image into a flat image so that the real topography of the surface can be appreciated. The profile parameters were recorded after applying a Gaussian filter, with an “end effect correction” activated.

2D and 3D images were also recorded from the virtual reconstruction made by the WLOI with a magnification of 20×.

#### 2.2.2. Roughness Parameters

The reported parameters were (Figure 1):
a.The arithmetical mean roughness (*R*_a_): the arithmetical mean height indicates the average of the absolute value along the sampling length.b.Root mean square deviation (*R*_q_): indicates the root mean square along the sampling length.c.Total height of profile (*R*_t_): indicates the vertical distance between the maximum profile peak height and the maximum profile valley depth along the evaluation length.d.Maximum height of profile (*R*_z_): indicates the absolute vertical distance between the maximum profile peak height and the maximum profile valley depth along the sampling length [27].


#### 2.2.3. Statistical Analysis

The arithmetical mean for the variables *R*_a_, *R*_q_, *R*_t_, and *R*_z_ of each implant system was calculated, and Fisher’s exact test was applied.

### 2.3. Evaluation of the Surface Chemical Composition

Surface chemical composition was evaluated using the electron spectroscopy for chemical analysis (ESCA), also known as X-ray photoelectron spectroscopy (XPS, ThermoFisher Scientific, Waltham, MA, USA). By the ESCA, x-ray irradiation hits on the sample surface and the detached energy is measured and analyzed. This method allows us to identify all the elements presented in the sample qualitatively and quantitatively, except for hydrogen (H) and helium (He) and generates a photoelectron spectrum, which includes characteristic peaks for all elements.

#### Data Acquisition

One area located in the middle part of the implant surface was randomly selected to evaluate the surface chemical composition. It was evaluated at two different stages:
Stage 1: as received by the manufacturer (AR), with a maximum time of two minutes of exposure to the air from the opening of the implant container until it was analyzed.Stage 2: After performing a 30-min sputter cleaning (SC) with an argon ion cannon, at an energy of 1 kV and a surface area of 1 mm^2^.

A K-Alpha^TM+^ XPS System (ThermoFisher Scientific, Waltham, MA, USA) was used for the surface chemical evaluation. The XPS was calibrated to irradiate the sample with a diameter spot of 400 microns, which represents an irradiation area of 0.35 mm^2^. An X-ray (monochromatic) photon cannon with an aluminum anode (1486.6 eV) was used, and atomic concentrations of each element were obtained from the areas under the peaks applying Scofield sensitivity factors. Data were registered and analyzed with Thermo Scientific^TM^ Avantage Data System Software (ThermoFisher Scientific, Waltham, MA, USA). A descriptive evaluation of the data obtained was carried out.

## 3. Results

### 3.1. Roughness of the Analyzed Surfaces. Quantitative Findings

The different roughness parameters were recorded in each of the nine implant systems, means for each parameter were calculated and are shown in Table 2; and maximum *R*_a_, minimum *R*_a_, mean *R*_a_, and standard deviation are shown in Table 3.

Group I: Straumann BL Implant System with SLA^®^ surface (obtained by a large grit sandblasting followed by acid-etching). The roughness parameters evaluated gave mean values of *R*_a_: 2.49 µm, *R*_q_: 3.16 µm, *R*_t_: 22.86 µm, and *R*_z_: 25.49 µm.

Group II: Microdent^®^ Implant System, with the surface treated with the Abrasive Treatment Extreme Cleaning^®^ (ATEC^®^) reported mean roughness values of *R*_a_: 1.07 µm, *R*_q_: 1.37 µm, *R*_t_: 15.01 µm, and *R*_z_: 28.09 µm, representing the second implant system with the least rough surface.

Group III: Astra Tech^®^ Implant System, with the OsseoSpeed^®^ surface, obtained after a titanium dioxide sandblasting at high pressure and followed by a fluoride nanostructure treatment, gave mean roughness values of *R*_a_: 1.97 µm, *R*_q_: 2.51 µm, *R*_t_: 24.71 µm, and *R*_z_: 35.05 µm.

Group IV: Avinent^®^ Implant System, with the Biomimetic^®^ surface, treated by sandblasting at high pressure, followed by addition of calcium and phosphorus, reported mean roughness values of *R*_a_: 2.39 µm, *R*_q_: 3.07 µm, *R*_t_: 27.58 µm, and *R*_z_: 35.95 µm, being the second implant system with the roughest surface.

Group V: Biomet 3i^®^ Implant System, with the Osseotite^®^ surface, treated by a double acid-etching technique, showed mean roughness values of *R*_a_: 0.79 µm, *R*_q_: 0.99 µm, *R*_t_: 17.07 µm, and *R*_z_: 29.74 µm, being the implant system with the smoothest surface.

Group VI: Klockner^®^ Implant System, with the Vega^®^ surface, which is treated by an alumina particles attack followed by a thermochemical treatment, reported mean roughness values of *R*_a_: 2.89 µm, *R*_q_: 3.74 µm, *R*_t_: 29.32 µm, and *R*_z_: 34.52 µm. This surface treatment provided the roughest surface of all evaluated implant systems.

Group VII: Mozograu-Ticare^®^ Implant System, with the RBM TC^®^ surface, treated by a sandblasting with resorbable particles followed by a double acid-etching, gave mean roughness values of *R*_a_: 1.31 µm, *R*_q_: 1.73 µm, *R*_t_: 36.71 µm, and *R*_z_: 59.30 µm.

Group VIII: Nobel Biocare^®^ Implant System, with the TiUnite^®^ surface, obtained after adding a thick and moderately rough titanium oxide layer, provided mean roughness values of *R*_a_: 1.10 µm, *R*_q_: 1.50 µm, *R*_t_: 24.21 µm, and *R*_z_: 32.12 µm.

Group IX: Galimplant^®^ Implant System, with the Nanoblast^®^ surface, treated with an abrasive sandblasting at high pressure followed by a triple acid-etching reported mean roughness values of *R*_a_: 1.45 µm, *R*_q_: 1.94 µm, *R*_t_: 14.98 µm, and *R*_z_: 16.38 µm.

### 3.2. Topography of the Analyzed Surfaces. Qualitative Findings

Qualitatively evaluating the surface topography features, a coincidence could be observed between the roughness pattern obtained in the 2D and 3D images from the WLOI at 20× and the classification of each implant system surface roughness after the *R*_a_ measurements. The greatest difference between peaks and valleys appeared in those groups of implant systems classified as “highly rough”, according to the *R*_a_. While a more smoothed roughness pattern was observed in those implant systems classified as “moderately and minimally rough” (Figure 2).

### 3.3. Surface Chemical Composition

Common elements, such as carbon (C), oxygen (O), and nitrogen (N), were found in all the samples analyzed at a high concentration, as these are found in the atmosphere. The highest average percentage of carbon was for group VIII (Nobel Biocare) with 82.4% (as received by the manufacturer, Kloten, Switzerland), followed by group III (Dentsply-Sirona), with 81.9%. The lowest average percentage of carbon was for group V (Zimmer Biomet) with 40.2%.

Titanium (Ti) also appeared as it is the main element of the alloys used in implantology. The concentration of these elements (C, O, and N) decreased in favor of the Ti concentration after the sputter cleaning (Table 4).

Aluminum (Al) was found in those implant systems that use a sandblasting with abrasive alumina oxide for the surface treatment (Microdent^®^, Avinent^®^, and Klockner^®^). In most of them, the percentage of Al increased after the sputter cleaning.

Elements such as silicon (Si), calcium (Ca), and sodium (Na) were found in small quantities in most of the samples evaluated, although their concentration decreased after sputter cleaning. Traces of elements such as chlorine (Cl), magnesium (Mg), phosphorus (P), or zinc (Zn) in small amounts were found in some of the samples evaluated. Most of them decreased or disappeared after sputter cleaning.

## 4. Discussion

The present study aimed to evaluate the relationship between the roughness created in different implant systems based on their surface treatment and the amount and type of trace elements found on their surface. Different roughness parameters and chemical elements found on the surface of the implants were evaluated.

Taking into account the different surface treatment techniques to achieve different topographical features, it is very important to characterize the surfaces and to study the surface chemical composition after each treatment. According to Wennerberg and Albrektsson [28], a light interferometer is a safe and effective way to measure the different roughness parameters. The white light optical interferometry allowed characterizing the 3D surface topography with a non-destructive method. The system was calibrated to a national institute of standards and technology (NIST) traceable standard, so any variation that could appear from one sample to another was excluded.

The implants’ surface microtopography is relevant at a cellular level in the osseointegration process. The nanotopography influences proteins/cells/implants interactions [23]. Nanotopography features induce changes at biological, physical, and chemical levels, causing an increase in the adhesion of osteogenic cells, and promoting osseointegration. It has been postulated that micro and nanosurfaces can influence osteoblastic activity and, therefore, osteoconduction process [29]. In this sense, some authors have shown an increase in bone-implant contact ratio (BIC) between four and ten weeks on surfaces with higher roughness. The current trend is to apply a surface treatment that generates the appropriate roughness to promote cell adhesion and bone neoformation [30,31].

The *R*_a_ values obtained in the present study were between 0.79 and 2.89 µm. Most commercial dental implants have a *R*_a_ value of 1–2 µm. This roughness range seems to be optimal for achieving osseointegration [12,32]. *R*_a_ values above 2 µm involve a prejudiced and non-reinforced bone response [28,32,33]. Our results showed three implant systems with a *R*_a_ value over 2 µm (highly rough). Two of these three implant systems received a surface treatment consisting of sandblasting without a subsequent acid-etching technique: group IV (Avinent^®^) and group VI (Klockner^®^). These surfaces could produce a faster osseointegration response, but the risk of future periimplantitis may be higher.

In the present study, five of the nine implant systems evaluated were ranged between the optimal *R*_a_ values, 1–2 µm (moderately rough). The two experimental groups that reached an ideal *R*_a_ value (near 1.5 µm) were group VII (Ticare^®^) and group IX (Galimplant^®^), which had received a surface treatment consisting of sandblasting followed by double or triple acid etching respectively. The combination of sandblasting, followed by some acidic etching, has been a technique widely used for surface treatment [34]. The reason for this combination is that by the sandblasting, an optimal roughness and mechanical properties are reached, while with the following acid etching, the peaks are smoothed and can add a high frequency component to the implant surface, which is of vital importance for protein adherence [34,35]. Some authors obtained a higher bone index contact with sandblasted and acid-etched surfaces when compared with other techniques like oxidized surfaces [36].

After the blasting deformation, some particles may become embedded and contaminate the implant surface [32]. By the use of the acid etching, the most superficial layers of the implant surface are removed, decreasing the surface stress and cleaning the surface contaminated by particles leftover from the sandblasting process. At the same time, this process helps in the creation of microcavities on the surface of the implant, generating an added nanometric roughness.

In the results of our study, only one experimental group presented a *R*_a_ value less than 1 µm, classifying itself as minimally rough (group V, Zimmer Biomet^®^). This may be because this implant system uses only a double acid etching as surface treatment, without previous blasting. As already known, blasting is the technique that allows achieving a greater surface roughness, being this roughness smoothed by the subsequent acid etching.

The residues or particles originated from the different processes used for surface treatment can remain attached to the surface of the titanium and cause some biological effects. These impurities coming from manufacturing or packaging processes may remain on the implant depending on the cleanliness process developed by the manufacturer [37].

In the present study, we evaluated the chemical surface composition with a XPS, which has been demonstrated to be a suitable technique to quantify the percentage of each chemical element present on the surface [38].

In our results, common elements (C, O, and N) were found in all samples analyzed at a high concentration. This fact is somewhat expected since these elements are found in the atmosphere, and the implant surface comes into contact with them. The highest average percentage of carbon was for group VIII (Nobel Biocare^®^), with 82.4% (as received by the manufacturer, Kloten, Switzerland). This high percentage may be due to the TiO_2_ layer surface treatment received by this group, which could be chemically attractive for the carbon union. The second-highest average percentage of carbon was for group III (Dentsply-Sirona^®^), with 81.9% (as received by the manufacturer, York, PA, USA), which surface is obtained after a titanium oxide sandblasting followed by a fluoride treatment. The existence of free fluoride anions on the surface could favor carbon union. The lowest average percentage of carbon was for group V (Zimmer Biomet^®^) with 40.2%, whose surface was obtained after a double acid etching technique. In all samples, the average percentage of carbon was decreased after sputter cleaning in favor of titanium.

The appearance of the oxygen was evident, since, in addition to being present in the atmosphere, it was part of the metallic oxides used during the sandblasting processes in most of the implant systems evaluated. The presence of OH groups is an important feature because it is strongly related to surface wettability, bioactivity, and protein adsorption [16,38]. The negative charge of the surface allows for a stronger electrostatic interaction with some proteins as albumin and fibronectin, enhancing the adhesion. This negative charge also attracts the water biomolecules (higher wettability) and ions like Ca_2+_ or PO_3−4_ (precipitation of hydroxyapatite) [39,40].

In our study appeared some surfaces that contained certain elements, such as Al, Si, Mg, P, or Zn. Such is the case for techniques that use sandblasting with aluminum oxide. During this process, some alumina particles may remain attached to the titanium surface. These particles influence free surface energy but are not heterogeneously distributed along the surface, and probably do not affect the distribution of fibronectin. However, they can affect protein absorption by their influence on wettability [1]. Some metals and metallic alloys release potentially harmful ions, such as Cr, Co, Ni, Al, or V, caused by wear of the implant at the bone-implant interface and the accumulation of metallic ions is considered to be the main reason for implant failure [41,42]. In our measurements, most of these elements drastically decreased or even disappeared after the sputter cleaning.

The present study possessed some strengths but also some limitations. All data has been recorded and calibrated by the same experimented operator, the experimental protocol was strictly developed, and non-destructive technology was utilized for the evaluation of the samples; on the other hand, the small sample size could compromise the data’s extrapolation.

The clinical relevance of the present work is that the topographical features of the implant surface are crucial for the proper osseointegration process. However, depending on the surface treatment technique used, traces of elements, metals, or ions may appear on the surface that may compromise the health of peri-implant tissues and develop future periimplantitis disease.

Further research on other implant surfaces, the development of new techniques for treating the surface, and different surface cleaning methods would be required to improve the results of this experimental study.

## 5. Conclusions

With the limitations of the present study, it could be concluded that the surface treatment technique influenced the roughness features. These roughness conditions, along with the physical–chemical characteristics of the technique used for conditioning the implant surface, affected the average percentage of the different elements on the implant surface. The presence of some of these elements might not be beneficial for peri-implant tissues. A better method of cleaning the final implant surface should be developed by the manufacturer before assembling the implant.

## Figures and Tables

**Figure 1 materials-13-00314-f001:**
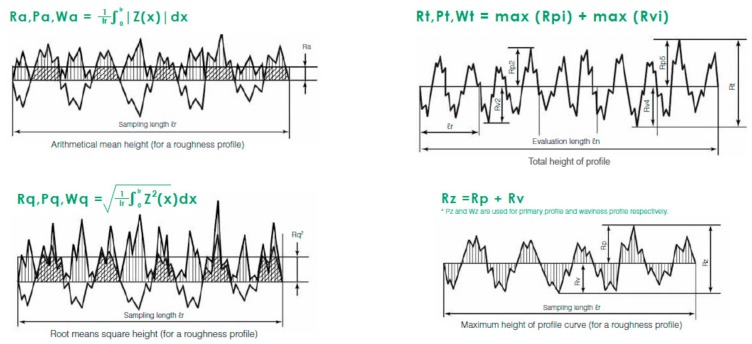
Graphics of the roughness parameters evaluated included *R*_a_, *R*_q_, *R*_t_, and *R*_z_ [27].

**Figure 2 materials-13-00314-f002:**
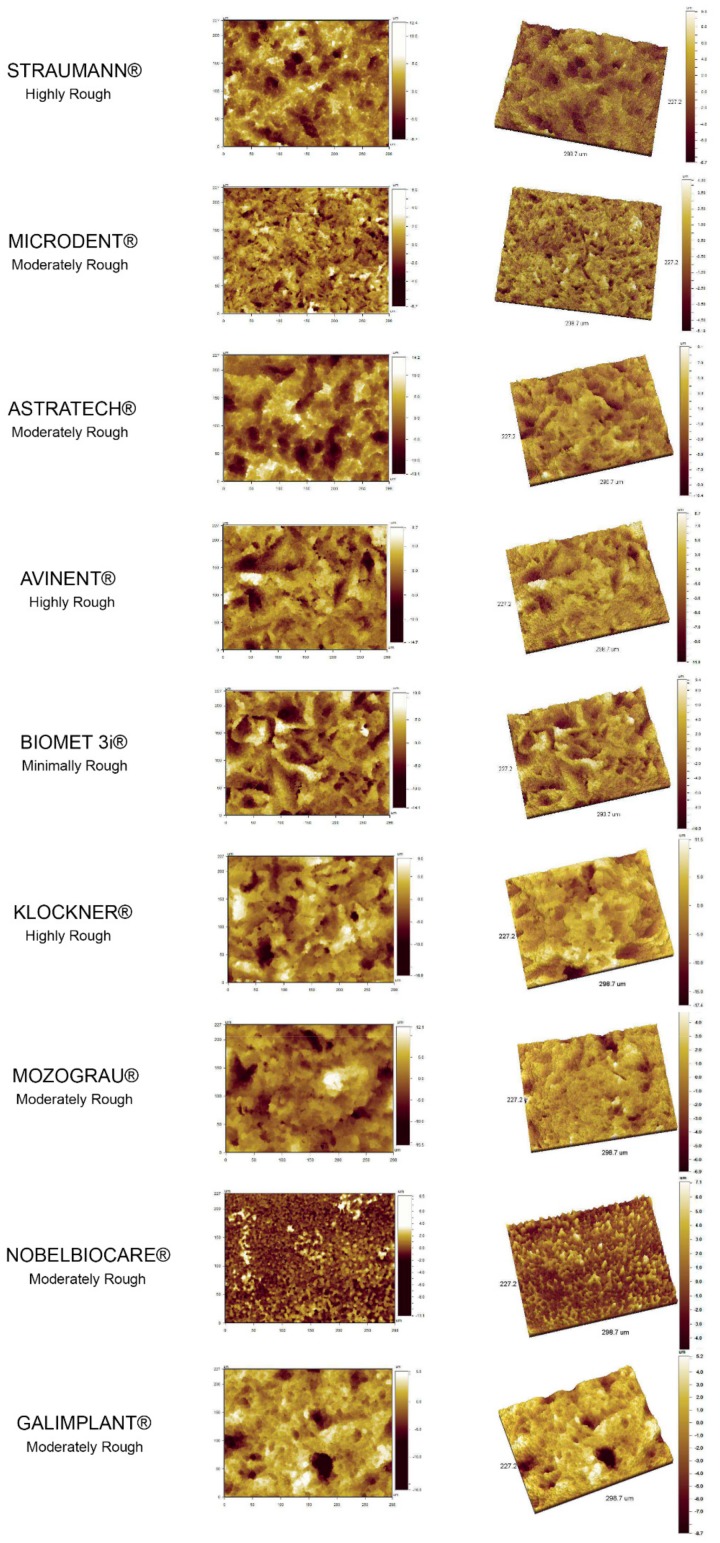
2D and 3D images of the topographic reconstruction of the implant surface for each of the nine implant systems evaluated, obtained from the white light optical interferometry (WLOI) with a magnification of 20×.

**Table 1 materials-13-00314-t001:** Implant manufacturer, implant system, surface name, surface treatment, and surface treatment code of each of the nine experimental groups.

Group	Implant Manufacturer	Implant System	Surface Name	Surface Treatment	Treatment Code
Group I	Straumann	BL	SLA^®^	large-grit sandblasting +	SB + AE
(*n* = 10)				acid-etching	
Group II	Microdent	Genius	ATEC^®^	Alumina sandblasting	SB (Al_2_O_3_)
(*n* = 10)					
Group III	Dentsply-Sirona	Astra Tech	OsseoSpeed^®^	Titanium Oxide sandblasting +	SB (TiO_2_) + F
(*n* = 10)				Fluoride treatment	
Group IV	Avinent	OCEAN^®^	Biomimetic^®^	Sandblasting +	SB + Ca + P
(*n* = 10)				Addition of calcium and Phosphorous	
Group V	Zimmer Biomet	Biomet 3i	Osseotite^®^	Double acid-etching	2AE
(*n* = 10)					
Group VI	Klockner	Vega^®^	Vega^®^	Alumina sandblasting +	SB (Al_2_O_3_) + TCT
(*n* = 10)				thermochemical treatment	
Group VII	Ticare	inHex^®^	RBM TC^®^	Resorbable particles sandblasting +	SB (res) + 2AE
(*n* = 10)				Double acid-etching	
Group VIII	Nobel Biocare	NobelReplace^®^	TiUnite^®^	TiO_2_ layer	TiO_2_
(*n* = 10)					
Group IX	Galimplant	IPX^®^	Nanoblast^®^	Sandblasting +	SB + 3AE
(*n* = 10)				Triple acid-etching	

**Table 2 materials-13-00314-t002:** Mean values of the roughness parameters evaluated for each implant system and the implant surface classification according to the degree of roughness.

Implant System	*R_a_* (µm)	*R_q_* (µm)	*R_t_* (µm)	*R_z_* (µm)	Roughness Type
STRAUMANN^®^	2.49	3.16	22.86	25.49	highly rough
MICRODENT^®^	1.07	1.37	15.01	28.09	moderately rough
ASTRATECH^®^	1.97	2.51	24.71	35.05	moderately rough
AVINENT^®^	2.39	3.07	27.58	35.95	highly rough
BIOMET 3i^®^	0.79	0.99	17.07	29.74	minimally rough
KLOCKNER^®^	2.89	3.74	29.32	34.52	highly rough
MOZOGRAU^®^	1.31	1.73	36.71	59.30	moderately rough
NOBELBIOCARE^®^	1.10	1.50	24.21	32.12	moderately rough
GALIMPLANT^®^	1.45	1.94	14.98	16.38	moderately rough

**Table 3 materials-13-00314-t003:** Sample number (n), minimum R_a_, maximum R_a_, mean R_a_, and R_a_ standard deviation of each experimental group.

Implant System	n	Minimum R_a_	Maximum R_a_	Mean R_a_	R_a_ Standard Deviation
STRAUMANN^®^	10	1.99	3.34	2.49	0.43126
MICRODENT^®^	10	0.95	1.13	1.07	0.05959
ASTRATECH^®^	10	1.80	2.14	1.97	0.10692
AVINENT^®^	10	1.99	2.78	2.39	0.27727
BIOMET 3i^®^	10	0.73	0.93	0.79	0.06567
KLOCKNER^®^	10	2.48	3.33	2.89	0.29578
MOZOGRAU^®^	10	1.06	1.47	1.31	0.11657
NOBELBIOCARE^®^	10	0.76	1.37	1.10	0.17735
GALIMPLANT^®^	10	1.07	1.72	1.45	0.24501

**Table 4 materials-13-00314-t004:** Percentage of each element found on the implant surface, as received from the manufacturer (AR) and after receiving a sputter cleaning (SC). “X” means than the element was not found in the sample.

System and Treat CODE	Stage	C	O	N	Ti	Al	Si	Ca	Na	Cl	Mg	P	Zn
STRAUMANN^®^	AR	58.4	28.4	1.2	7.0	X	3.5	0.8	0.8	X	X	X	X
SB+AE	SC	23.8	47.8	0.9	25.5	X	1.8	0.2	X	X	X	X	X
MICRODENT^®^	AR	54.7	28.0	2.5	4.8	5.1	1.8	0.5	1.3	0.7	0.7	X	X
SB (Al_2_O_3_)	SC	23.9	41.6	0.9	17.5	15.3	X	0.1	X	X	X	X	X
ASTRATECH^®^	AR	81.9	13.8	X	1.8	X	1.0	1.4	X	X	X	X	X
SB (TiO_2_)+F^-^	SC	47.7	21.3	2.2	28.6	X	X	0.0	X	X	X	X	X
AVINENT^®^	AR	45.5	36.5	1.1	9.7	3.3	1.6	1.7	X	X	X	0.6	X
SB+Ca+P	SC	41.4	38.8	1.1	11.2	1.4	2.6	2.8	X	X	X	0.7	X
BIOMET 3i^®^	AR	40.2	39.6	1.2	11.8	X	7.2	X	X	X	X	X	X
2AE	SC	38.2	27.5	2.3	32.1	X	X	X	X	X	X	X	X
KLOCKNER^®^	AR	62.6	23.6	1.1	2.9	4.3	4.8	0.4	X	X	X	0.5	X
SB (Al_2_O_3_)+TCT	SC	64.1	18.3	X	7.3	8.5	1.7	X	X	X	X	X	X
MOZOGRAU^®^	AR	73.5	18.2	0.4	2.7	X	2.3	1.3	0.7	X	X	1.0	X
SB (res)+2AE	SC	47.7	21.3	2.2	28.6	X	X	0.2	X	X	X	X	X
NOBELBIOCARE^®^	AR	82.4	12.2	X	0.7	X	2.0	1.9	X	X	X	0.8	X
TiO_2_	SC	69.9	18.8	X	8.5	X	1.7	0.8	X	X	X	2.0	X
GALIMPLANT^®^	AR	53.1	31.2	1.0	8.4	X	3.7	1.2	X	X	X	X	0.50
SB+3AE	SC	33.7	27.1	3.7	31.5	X	X	X	X	X	X	X	X

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
