# Peer review of "Influence of the Titanium Implant Surface Treatment on the Surface Roughness and Chemical Composition"

_materials, 2020, doi:10.3390/ma13020314_

Round 1

Reviewer 1 Report

The manuscript entitled "Influence of the Titanium Implant Surface Treatment
on the Surface Roughness and Chemical Composition" was well designed and written well.

few minor comments to be addressed:

Preparation of the material the procedures have to be written in detail, directly shifting into the groups will confuse the readers. how come the author comes to the conclusion only with surface roughness and TEM analysis. In the introduction author was discussing the elements, the author has to perform either EDX or any other elemental analysis to add move values to his statements.     The author has mentioned in the abstract " the surface treatment influences the roughness features and affect the average
36 percentage of the elements on the implant surface" justify this statement with results. Did the author performed any biocompatibility analysis to show how the cells adhere or it's biocompatible and compatible with the tissue?

Author Response

REVIEWER 1.

Dear Reviewer.

Thank you for your careful reading and review of this manuscript.

We will proceed with the clarifications of the comments made in your review point by point:

REVIEWER

Preparation of the material the procedures have to be written in detail, directly shifting into the groups will confuse the readers.

AUTHORS

A new paragraph has been written (lines 98-100) as an introduction before describing experimental groups.

REVIEWER

how come the author comes to the conclusion only with surface roughness and TEM analysis.

AUTHORS

Our results showed different roughness values in the different groups (treated by different conditioning procedures) and different surface chemical composition, that is why we conclude that the surface treatment influences roughness and chemical composition.

REVIEWER

In the introduction author was discussing the elements, the author has to perform either EDX or any other elemental analysis to add move values to his statements.   

AUTHORS

Several authors conclude that the Electron Spectroscopy for Chemical Analysis (ESCA), or X-ray Photoelectron Spectroscopy (XPS) is a valid method to analyze surface chemical composition.

REVIEWER

 The author has mentioned in the abstract " the surface treatment influences the roughness features and affect the average percentage of the elements on the implant surface" justify this statement with results.

AUTHORS

As answered before in point 2, our results showed differences in both roughness and the average percentage of the elements.

REVIEWER

Did the author performed any biocompatibility analysis to show how the cells adhere or it's biocompatible and compatible with the tissue?

AUTHORS

No, we didn’t as it wasn’t the goal of the present study. Some authors have already performed similar analyses. We think it’s a very interesting issue and we don’t discard it in future studies.

Reviewer 2 Report

I read with great interest the manuscript entitled: "Influence of the Titanium Implant Surface Treatment on the Surface Roughness and Chemical Composition" written by Ana I. Nicolas-Silvente and co-workers.
the article discuss the surface roughness and the surface treatment chosen by the company. In the second part, the surface residues of other materials on the surface itself are evaluated, related to the cleaning method. the authors conclude that "The presence of these elements may not be beneficial for peri-implant tissues. The cleaning process at the implant surface should be improved by the manufacturer before assembling the implant. "
although the topic may be interesting i think it is not ready for publication for different reasons.

in general, the surface roughness relating to the different surface treatments is already known. very often the companies themselves provide this information. the goal of this study is to relate it, to the traces of other materials present. in this case too, the companies supply the exact percentages of the surface residues present. therefore the goal of the study is inconsistent and should be changed to make the paper interesting.
basically, the real point of interest could be the presence of traces with and without surface cleaning but this has not been sufficiently treated.
A statistical relationship between the variables studied has not been established, but only between the different roughnesses. As mentioned, roughnesses are already known topics. Interest may have shifted more to the second part of the study.

Beware of formatting.
In references, the volume is italicized.
The author name must be pointed. not all of them are.

Author Contributions goes with the initials of the authors, not full names.
in the header, the matching author must be placed after the affiliation number and with a comma. I recommend putting initial as suggested in the template.

abstract

miss aim and check the number of words.
abbreviations in the abstract may exist but must be explained.
the results in the abstract are all listed together. the groups should be divided (at least between ar and sc) to understand if there are differences

line 36 "The presence of these elements may not be beneficial for peri-implant tissues."
These conclusions are not relevant to the data collected. can be put in the discussion but not in the conclusions of the abstract. are not concluded drawn from the data collected.

intro
from line 69 to 75 there are no references to what has been said. should be added

line 83 LDH. must be explained. what is it? not just abbreviation

a part is missing which introduces the cleaning methods and more specifically the "sputter cleaning" mentioned by the authors.
line 83 This toxicity, together with the ionic release, could somehow affect the osseointegration process and perhaps could be the precursor of a future peri-implant disease [23]. "
the reference refers to the cytotoxicity of the anesthetic. is not relevant

materials and methods
different subheadings title are not formatted in the same way
subheadings 3.1.1 3.2.2 et al should be written as the same paragraph. they explain the same result in different group.
if averages are presented, standard deviation and minimum and maximum should be present.

line 264 "3.2. Topography of the analyzed surfaces. Qualitative Findings. "
this paragraph is not useful. adds nothing to the study results. clearly if there are surface variations (measured with different parameters) the qualitative analysis of the images will reflect the same information.

image 2 no axes are read. the image is of poor quality

a paragraph on statistical analysis is not present. the sample size must be added, the statistical methods explained.

discussion
line 312 "Albrecktsson" is not well written.

line 376 "In all samples, the average percentage of carbon was decreased after sputter cleaning in favor of titanium." this is the most interesting data. everything should be redone according to this variable. in my opinion the most interesting.

part of the discussion and conclusions are not relevant. the authors cannot speak of benefits for peri-implant tissues. it is not an objective of the study.

Author Response

REVIEWER 2.

Dear Reviewer.

Thank you for your careful reading and review of this manuscript.

We will proceed with the clarifications of the comments made in your review point by point:

REVIEWER

in general, the surface roughness relating to the different surface treatments is already known. very often the companies themselves provide this information. the goal of this study is to relate it, to the traces of other materials present. in this case too, the companies supply the exact percentages of the surface residues present. therefore the goal of the study is inconsistent and should be changed to make the paper interesting.
basically, the real point of interest could be the presence of traces with and without surface cleaning but this has not been sufficiently treated.

AUTHORS

Although the companies offer us information about the roughness and surface chemical composition, this study aimed to establish a relationship between surface treatment, the roughness created, and the amount of trace elements (surface contamination). The use of different techniques for surface cleaning is a very interesting topic that we hope to be able to develop soon, following this research line in future studies.

REVIEWER
A statistical relationship between the variables studied has not been established, but only between the different roughnesses. As mentioned, roughnesses are already known topics. Interest may have shifted more to the second part of the study.

AUTHORS

Indeed, only a statistical study of the roughness variables has been done. In the second part of the study (surface chemical evaluation), just a descriptive study has been done, describing the average percentage of each chemical element. This data does not allow us to make another type of statistic as the goal was not to compare between groups. We could have evaluated the intragroup differences at the two times: as received by the manufacturer and after the sputter-cleaning to evaluate in which roughness parameter is more effective this type of cleaning, but it wasn’t the goal of this study.

REVIEWER

Beware of formatting. In references, the volume is italicized. The author name must be pointed. not all of them are. Author Contributions goes with the initials of the authors, not full names.in the header, the matching author must be placed after the affiliation number and with a comma. I recommend putting initial as suggested in the template.

AUTHORS

Thank you for writing down these details. All the “formatting” issues have been corrected, as suggested by the reviewer.

REVIEWER

abstract

miss aim and check the number of words.abbreviations in the abstract may exist but must be explained. the results in the abstract are all listed together. the groups should be divided (at least between ar and sc) to understand if there are differences

AUTHORS

The aim was added to the abstract. The number of words is now 194 of a maximum of 200. All abbreviations have been deleted or explained. The results have been divided, as suggested.

REVIEWER

line 36 "The presence of these elements may not be beneficial for peri-implant tissues."
These conclusions are not relevant to the data collected. can be put in the discussion but not in the conclusions of the abstract. are not concluded drawn from the data collected.

This sentence has been removed from the abstract and is argued in the discussion section, as suggested by the reviewer.

REVIEWER

intro
from line 69 to 75 there are no references to what has been said. should be added

AUTHORS

A new quote has been added: [20] Marenzi, G.; Impero, F.; Scherillo, F.; Sammartino, J.C.; Squillace, A.; Spagnuolo, G. Effect of different surface treatments on titanium dental implant micro-morphology. Materials. 2019, 12, 733.

REVIEWER

line 83 LDH. must be explained. what is it? not just abbreviation

AUTHORS

The meaning of DNA: deoxyribonucleic acid and LDH: lactate dehydrogenase, has been described in the text.

REVIEWER

a part is missing which introduces the cleaning methods and more specifically the "sputter cleaning" mentioned by the authors.

AUTHORS

A new paragraph has been added, introducing cleaning methods.

REVIEWER
line 83 This toxicity, together with the ionic release, could somehow affect the osseointegration process and perhaps could be the precursor of a future peri-implant disease [23]. "the reference refers to the cytotoxicity of the anesthetic. is not relevant

AUTHORS

This reference has been changed.

REVIEWER

if averages are presented, standard deviation and minimum and maximum should be present.

AUTHORS

A new table has been inserted as “Table 3” with the requested data.

REVIEWER

image 2 no axes are read. the image is of poor quality

AUTHORS

By presenting a high number of images assembled all together, the quality and size of them are diminished. We can send these images separately if the publisher estimates it.

REVIEWER

a paragraph on statistical analysis is not present. the sample size must be added, the statistical methods explained.

AUTHORS

Both are explained in material and methods and in “table 3”.

REVIEWER

discussion
line 312 "Albrecktsson" is not well written.

AUTHORS

Change to "Albrektsson" is done.

Reviewer 3 Report

The manuscript “Influence of the Titanium Implant Surface Treatment  on the Surface Roughness and Chemical Composition” aimed to evaluate the relationship between the roughness created in different implant systems based on their surface treatment and the amount and type of trace elements found on their surface.

The study is interesting and may be published in Materials journal.

However, some minor corrections should be performed for example:

The aspects concerning the impurities located on the implant surface and their role in the occurrence of peri-implantitis and foreign bodies reaction could be mentioned in the discussions paragraph (Duddeck, D. U., Albrektsson, T., Wennerberg, A., Larsson, C., & Beuer, F. (2019). On the Cleanliness of Different Oral Implant Systems: A Pilot Study. Journal of clinical medicine, 8(9), 1280).)

Several corrections concerning the decimal separator need to be done: sometimes is a coma (ex. abstract, rows 280, 281, 282…, Table 2, ) and sometimes is a point (rows 60, 61…). Same confusion could be found in whole manuscript.

Author Response

REVIEWER 3.

Dear Reviewer.

Thank you for your careful reading and review of this manuscript.

We will proceed with the clarifications of the comments made in your review point by point.

The aspects concerning the impurities located on the implant surface and their role in the occurrence of peri-implantitis and foreign bodies reaction could be mentioned in the discussions paragraph (Duddeck, D. U., Albrektsson, T., Wennerberg, A., Larsson, C., & Beuer, F. (2019). On the Cleanliness of Different Oral Implant Systems: A Pilot Study. Journal of clinical medicine, 8(9), 1280).)

We consider this new bibliography quote very interesting, so we include it in our discussion.

Several corrections concerning the decimal separator need to be done: sometimes is a coma (ex. abstract, rows 280, 281, 282…, Table 2, ) and sometimes is a point (rows 60, 61…). Same confusion could be found in whole manuscript.

Corrections have been done turning the decimal separator “coma” for “point”.

Round 2

Reviewer 2 Report

The reviewer would like to thank the authors for addressing most of the comments. The manuscript has improved. However, I still have some concerns minor errors.

Authors inserted a paragraph on cleaning methods. I think it appropriate to extend the part on sputter cleaning. so we can better introduce this study variable.
to better understand (from a graphic point of view)materials and methods I suggest the authors to eliminate paragraphs 2.1.1 and others. inserting the groups in a single paragraph. the same for the different results 3.1.1. and similar.

Author Response

The authors would like to thank the reviewer for the kind comments.

Regarding the “cleaning methods” we have added some information about sputter cleaning in the introduction (lines 89-95).

Paragraphs 2.1.1 to 2.1.9, as well as 3.1.1 to 3.1.9 has been eliminated, inserting groups in a single paragraph as suggested.
